# Liquid Overlay-Induced Donor Plant Vigor and Initial Ammonium-Free Regrowth Medium Are Critical to the Cryopreservation of *Scrophularia kakudensis*

**DOI:** 10.3390/plants13172408

**Published:** 2024-08-28

**Authors:** Hyoeun Lee, Hana Park, Sang-Un Park, Haenghoon Kim

**Affiliations:** 1Department of Agricultural Life Science, Sunchon National University, Suncheon 57922, Republic of Korea; yee0430@scnu.ac.kr (H.L.); qkrgksk1102@naver.com (H.P.); 2Department of Crop Science, Chungnam National University, Daejeon 34134, Republic of Korea

**Keywords:** A3-80%, droplet-vitrification, endangered species, gelling agent, subculture medium, three-step regrowth, two-step preculture

## Abstract

Cryopreservation, storing biological material in liquid nitrogen (LN, −196 °C), offers a valuable option for the long-term conservation of non-orthodox seeds and vegetatively propagated species in the sector of agrobiodiversity and wild flora. Although the large-scale cryobanking of germplasm collections has been increasing worldwide, the wide application of cryopreservation protocols in wild flora is hampered by difficulties in vitro propagation and a lack of universal cryopreservation protocols, among others. This study established a systematic approach to developing an in vitro culture and droplet-vitrification cryopreservation procedure for shoot tips of *Scrophularia kakudensis*. The standard procedure includes a two-step preculture with 10% sucrose for 31 h and with 17.5% sucrose for 16 h, osmoprotection with loading solution C4-35% (17.5% glycerol + 17.5% sucrose, *w*/*v*) for 30 min, cryoprotection with A3-80% (33.3% glycerol + 13.3% dimethyl sulfoxide + 13.3% ethylene glycol + 20.1% sucrose, *w*/*v*) at 0 °C for 60 min, and cooling and rewarming using aluminum foil strips. After unloading, a three-step regrowth procedure starting with an ammonium-free medium with growth regulators was essential for developing normal plantlets from cryopreserved shoot tips. Liquid overlay on the gelled medium two weeks after inoculation resulted in vigorous growth during subcultures. Moreover, liquid overlay increased LN regeneration by up to 80%, i.e., 23% higher than no liquid overlay.

## 1. Introduction

*Scrophularia kakudensis*, Scrophulariaceae Family, is an endangered species in Korea, and this genus is pharmaceutically important against inflammation and gastrointestinal problems [1,2]. In vitro adventitious shoots were induced from the nodal explants containing acacetin (a flavonoid), phenol, and flavonoid with free radical scavenging ability [3]. 

Combined with in vitro technologies, cryopreservation, storing biological material in liquid nitrogen (LN, −196 °C), offers a unique option for the long-term conservation and restoration of non-orthodox or limitedly available seeds and vegetatively propagated species [4,5,6]. However, the cryopreservation of endangered species often encounters challenges due to the lack of in vitro propagation systems and standard cryopreservation protocols and the scarcity of plant material [6,7]. Challenges faced in establishing an in vitro propagation system include microbial contamination, poor growth, etc. Hence, in vitro propagation and cryopreservation success is often case by case [4]. A guideline for the successful cryopreservation of clonally propagated plants has frequently been considered at least 20–40% regeneration due to the lower levels of the initial regeneration percentage of plant cryopreservation [8], which limits the large-scale cryobanking of phyto diversity in the era of climate change.

As a solution-based vitrification technique, droplet-vitrification (DV) is a multi-stage procedure with several factors from the stage (1), involving material preparation, to (2), implementing the protocol {pre-LN (preculture, osmoprotection, cryoprotection with vitrification solution), LN (cooling, rewarming, unloading)}, and (3), regrowth. However, most literature has focused on the protocol, the main body of cryobiotechnology, and spontaneously tested the choices of explant [9] and regrowth medium [10]. Plant materials are prepared by the in vitro propagation of donor plants under an established procedure, depending on the species and type of organ. Diverse options, such as media, growth regulators, gelling agents, environmental conditions, and critical manipulation skills, may affect their growth. 

Like seed vigor [11], donor plant vigor (DPV) for cryopreservation may refer to the donor plant properties that determine the ability to regrow quickly and uniformly and regenerate into normal plantlets under a wide range of cryopreservation conditions and procedures [12]. Rapid growth and eventually enormous dry weight with no or little growth hormones can indicate DPV since the explants from the higher DPV can tolerate the cytotoxicity of plant vitrification solutions (PVS), resulting in higher LN regeneration [13]. 

In this study, we aimed to establish an in vitro propagation system for *S. kakudensis* and develop a DV procedure for cryopreservation. This study highlights the critical role of liquid overlay-induced donor plant vigor in the material preparation stage and initial ammonium-free regrowth medium in the regrowth stage. 

## 2. Results

### 2.1. In Vitro Propagation of Scrophularia kakudensis

#### 2.1.1. Effect of Subculture Medium and Conditions

Nodal segments were cultured in seven condition variants (standard + six alternative conditions) for six weeks. One lamp (Lamp1, 40 µE m^−2^ s^−1^), instead of two lamps (standard, 60 µE m^−2^ s^−1^), significantly increased the shoot length but not the dry weight (Figure 1). It increased the size and dry weight of the roots, too. The growing pattern of half-strength hormone-free MS medium (1/2MSF) was similar to the standard condition of full-strength hormone-free MS (MSF). 

Other conditions, such as growth hormones of 0.5 mg L^−1^ each of benzyl adenine (BA) and α-naphthaleneacetic acid (NAA) (BA + NAA0.5), activated charcoal (AC1g), agar (Agar8g), or Gellan gum + agar (ge1.5 + Ag4), resulted in an inferior length and dry weight compared to the standard condition (full strength hormone-free MS + Gellan gum 3.0 g L^−1^, two lamps).

Based on the results, the standard subculture condition for *S. kakudensis* node cuttings was set as full-strength hormone-free MS medium (MSF) supplemented by Gellan gum 3.0 g L^−1^, and provided with one fluorescent lamp (40 µE m^−2^ s^−1^) for six weeks. 

#### 2.1.2. Effect of Liquid Overlay on Top of the Gelled Medium

We examined the effect of the liquid overlay on the in vitro growth during subcultures. Liquid MSF medium was overlayed (LO) on top of the Gellan gum-gelled medium at the time of node section inoculation (W0), after one week (W1), two weeks (W2), or no overlay (LX). After six weeks, the height and dry weight of subcultured plantlets overlayed (W0, W1, and W2) were significantly higher than non-overlayed (Liq-X): shoot length was 3.8 cm → 9.4–9.8 cm and shoot dry weight was 3.9 mg → 13.4–16.2 mg (Figure 2 and Figure 3). The shoot dry weight for LO(W2) (16.2 mg) was heavier than LO(W0) (13.4 mg) or LO(W1) (3.9 mg), with no significant difference in plant height.

However, overlaying at the time of inoculation (W0) or after one week (W1) induced symptoms of hyperhydricity: the leaves looked wet. The hyperhydricity symptoms were more frequent in the order of LO((W0) > LO(W1) > LO(W2). In contrast, no liquid overlay (LX) did not show the symptoms.

#### 2.1.3. Effect of Liquid Overlay Composition (Nutrients and Sucrose)

We further investigated the inclusion or exclusion of MS nutrients and sucrose in the Gellan gum-gelled medium and liquid overlay on the plant height and dry weight of subcultured plantlets to elucidate the mechanism of the liquid overlay. 

When the Gellan gum-gelled medium was filled with the MS nutrients (macro, micro, amino acids, and vitamins) and 3% sucrose {G(+N+S)}, the height and dry weight of shoots and roots in liquid-overlayed plantlets were higher regardless of the components of liquid {+L(+/−N, +/−S)} if only the liquid were overlayed: shoots were 9.2–9.8 cm, 9.9–11.9 mg and roots were 4.8–6.2 cm, 2.8–3.2 mg (Figure 3). Even the overlay of distilled water {G(+N+S)+L(−N−S)} supported vigorous growth of donor plantlets. 

Sucrose omitted-Gellan gum-gelled MS medium resulted in marginal growth even with normal MS liquid overlay {G(+N−S)+L(+N+S)}: shoots were 2.5 cm, 1.6 mg and roots were 1.2 cm, 0 mg. When sucrose was omitted in both gelled and liquid medium {G(+N−S)+L(+N−S), G(+N−S)+L(−N−S)}, plant height and dry weight were nil (shoots were 0.7–0.9 cm, 0.2–0.3 mg, and roots were 0 cm, 0 mg), implying that photo-autotrophic performance of the plantlets was marginal. 

This result indicates that in vitro plantlets primarily uptake nutrients and sucrose from the gelled medium. The liquid overlay significantly facilitated the uptake regardless of the liquid composition; distilled water was equally effective. Hence, an overlay of distilled water per se enables the intake of nutrients and sucrose from the gelled medium, possibly by modifying the osmotic potential of the gelled medium and plantlets. All four sucrose-omitting gelled mediums {G(+N−S)} produced marginal dry weights. 

**Figure 3 plants-13-02408-f003:**
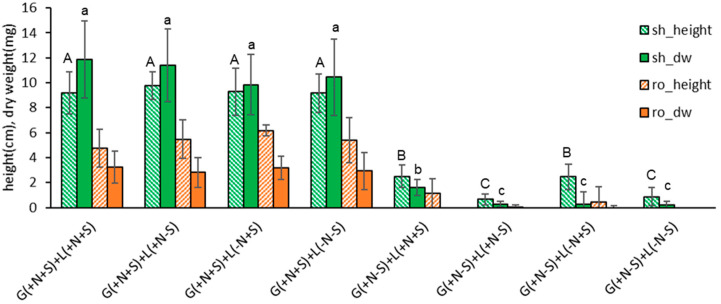
Effect of gelled medium and liquid overlay components on plant height (cm) and dry weight (mg) of subcultured *Scrophularia kakudensis* plantlets. G, Gellan gum gelled; L, liquid overlay; and inclusion (+) or exclusion (−) of nutrients (N) and sucrose (S). Means with the same letters (A–C and a–c) in each column are not significantly different by least significant difference test (LSDT, *p* < 0.05).

A similar pattern was noticed in the dry weight to fresh weight ratio (DW/FW). When the Gellan gum-gelled medium was filled with nutrients and sucrose {G(+N+S)}, the DW/FW of shoots and roots in liquid overlayed plantlets was higher than the gelled medium without sucrose {G(+N−S)}: shoots 0.098 and roots 0.080 vs. shoots 0.050 and roots 0.002 on average (Figure 4). The DW/FW of LX in Figure 2 was 0.099 for shoots and 0.077 for roots, respectively, so liquid overlayed plantlets were subjected to normal metabolism. However, when sucrose was omitted in both gelled and liquid medium {G(+N−S)+L(+N−S), G(+N−S)+L(−N−S)}, the lowest shoot DW/FW (0.033–0.039) was noticeable, reflecting the marginal photo-autotrophic property.

### 2.2. Cryopreservation of Scrophularia kakudensis Shoot Tips

#### 2.2.1. Pre-LN Stages in Droplet-Vitrification Procedure

Using the temporary standard procedure, we tested diverse options, such as sucrose preculture, no-osmoprotectant, cooling/rewarming devices, and regrowth medium. A step-wise preculture with 10% sucrose (S-10%) for 31 h and 17.5% sucrose (S-17.5%) for 17 h produced higher post-cryopreservation (LN) regeneration (83.3%), which was higher than the one-step preculture with S-10% for two days (S-10%, 76.3%) (Figure 5A,B). In contrast, higher sucrose concentrations of S-10% → S-25% (S-25%) resulted in the lowest LN regeneration of 66.4% due to the osmotic stress. 

No osmoprotection (no-OP) was stressful to the shoot tips and eventually resulted in lower LN survival and regeneration (42% and 30% lower) compared to the standard condition of OP-treated shoot tips. When the cryovial (vial) was used as a cooling device instead of aluminum foil strips, possibly insufficient cryoprotection with A3-80% resulted in lower LN survival and regeneration (28% and 30% lower) due to the lower cooling/warming velocity than the aluminum foil strips. A conventional one-step ammonium-containing regrowth medium and growth regulators (RM2~) produced lower LN regeneration (45% lower) compared to the standard three-step regrowth with an initially ammonium-free medium (RM1-2-F). A three-step regrowth procedure starting with an ammonium-containing medium followed by the same medium and finally without growth regulators (RM2-2-F) produced 39% lower LN than the standard condition. This result suggests that ammonium ion during the initial five days is toxic to the cryopreserved shoot tips, and transferring to a new ammonium-containing medium twice has a marginal effect.

Among the conditions tested, the composition of the initial regrowth medium (with or without ammonium ion) and steps are more critical factors than any conditions tested in pre-LN stages. This result implies that the DV procedure, especially cryoprotection with A3-80% and cooling and rewarming, was stressful to the shoot tips, possibly resulting in ammonium-induced ROS-mediated oxidative stress during regrowth stages. However, this should be confirmed by extensive analysis. In addition, the combined effect of ammonium-free medium and growth regulators during three-step regrowth stages needs further investigation.

The PVSs significantly affected the LN recovery of the shoot tips (Figure 6A,B). Among the PVSs tested, A3-80% produced the highest LN regeneration (83.3%, 18% higher than PVS2). The *S. kukudensis* shoot tips were sensitive to the cytotoxicity, especially chemical toxicity, of A3-90% and the osmotic stress of PVS3 (B1-100%). Hence, a diluent of A3-90%, A3-80% (21% higher), and a diluent of PVS3, B5-85% (10% higher) produced higher LN regeneration than the original PVSs. 

#### 2.2.2. Three-Step Regrowth in Droplet-Vitrification Procedure

We investigated the combined effects of ammonium ion presence or absence (+ or −) and plant growth hormones (GA_3_ + BA) in the regrowth medium at steps 1, 2, and 3 for 5, 23, and 14 days, respectively. The specific combinations of factors were represented as step 1/step 2/step 3, as illustrated in Figure 7. 

The most critical conditions for LN survival and regeneration were omitting ammonium and adding growth hormones in step 1. The condition for higher LN survival (77.1–84.6%) and regeneration (72.9–82.1%) was ammonium-free medium with growth hormones {1 mg L^−1^ gibberellic acid (GA_3_) and 0.5 mg L^−1^ benzyl adenine (BA)} in step 1 (−a + h/). The presence or absence of growth hormones in steps 2 and 3 was less important. Other treatments, i.e., either ammonium-containing or the absence of growth hormones (+a or −h/) in step 1, resulted in lower LN survival (31.3–50.0%) and regeneration (16.3–22.7%), regardless of step 2 and 3 conditions. 

This result reflects ammonium ion toxicity and growth hormones’ beneficial effect in step 1. We found that combining an initial ammonium-free medium and growth hormones during the regrowth stage (step 1) had a more pronounced impact on the outcomes than any pre-liquid nitrogen (pre-LN) stages or conditions examined in Figure 5 and Figure 6. This result suggests that the specific composition of the regrowth medium at the initial stage plays a crucial role in determining the success of post-cryopreservation survival and regeneration.

#### 2.2.3. Effect of Liquid Overlay (Timing and Composition) in Droplet-Vitrification Procedure

Since the liquid overlay on top of the Gellan gum-gelled MSF medium significantly increased plant height and dry weight during the subculture, we investigated the effect of liquid overlay on subsequent LN regeneration. 

With the liquid overlay at the time of node section inoculation {LO(W0), G(+N+S)+L(N+S)}, hyperhydricity symptoms (wet leaves) were observed in about one-third of the in vitro plants (calculated in Table 1 as 31.3%). Hence, only normal plantlets were subjected to cryopreservation via the DV procedure, resulting in slightly increased LN regeneration compared to no liquid overlay {LX, G(+N+S) −L(−N−S)} (56.8% → 64.2%) (Figure 8). Liquid overlay after two weeks of inoculation {LO(W2), G(+N+S)+L(+N+S)} significantly increased the regeneration of cryoprotected (LNC) and cryopreserved (LN) shoot tips compared to no liquid overlay (LNC 62.1% → 93.8%, LN 56.8% → 80.0%). 

Interestingly, the liquid overlay of distilled water {LO(W2), G(+N+S)+L(−N−S)} also slightly increased the LN regeneration (56.8% → 63.3%), but it was significantly lower than the liquid overlay of the standard liquid medium at week 2 {LO(W2), G(+N+S)+L(+N+S)}. Omitting sucrose in Gellan gum-gelled medium {LO(W2), G(+N−S)+L(+N+S)} resulted in similarly marginal LNC and LN regeneration (20%). Hence, the lower LN regeneration is attributed to PVS-induced cytotoxicity during cryoprotection rather than ice crystallization during cooling and warming. 

The effect of liquid overlay on the in vitro growth (Figure 2 and Figure 3) and subsequent LN regeneration (Figure 8) was combined in Figure 9. Liquid overlay timing and its composition impacted the growth of *S. kukudensis* plantlets during subculture and subsequent LN regeneration. Liquid overlay increased shoots’ and roots’ length and dry weight, eventually determining the LN regeneration. The LN regeneration positively correlated with shoot height (*r* = 0.943), shoot dry weight (*r* = 0.973), and root dry weight (*r* = 0.885) among the treatments, which supports that donor plant vigor is a prerequisite for cryopreservation. 

The percentage of wet leaves (hyperhydricity) caused by the liquid overlay was detrimental to cryoprotection with PVS and, eventually, cryo-exposure. Hence, LN regeneration negatively correlated with the hyperhydricity percentage (*r* = −0.796) of the leaves during the subculture. 

We dug into the three liquid overlay treatments (WO, W2, and LX) during the subculture and observed hyperhydricity at 31.3%, 4.2%, and 0%, respectively (Table 1). These normal and hyperhydric plantlets were subjected to node cutting for shoot-tip sampling of cryopreservation. 

At week two, nodal sections sprouted two axillary shoots; one grew faster (primary) than the other (secondary) at week three. At week four, the primary shoots reached 4.0–6.5 cm long, and the secondary shoots reached 1.5–2.5 cm (Table 1, Figure 10A,C). These node-induced shoots also show symptoms of hyperhydricity from 0 to 100% (Table 1, Figure 10A), depending on the liquid overlay timing in the last subculture and the primary or secondary sprouting of axillary buds. For instance, in the liquid overlay at week two {LO(W2)}, 95.8% of normal subcultured plantlets sprouted primary and secondary shoots: primary shoots (6.5 cm long) have no hyperhydricity, and secondary shoots have 25% hyperhydricity on sprouted shoots. A total of 4.2% of hyperhydrated plantlets at week 2 sprouted 100% hyperhydric shoots in node culture. 

After following cryopreservation using these node-section-induced primary and secondary shoots, we determined the LN regeneration of 80% (normal-primary, Figure 10D), 50% (normal-secondary), 60% (hyperhydric-primary, Figure 10B), and 35% (hyperhydric-secondary). The hyperhydric-secondary shoots at weeks 0 (W0) and 2 (W2) produced 20% and 35% LN regeneration.

Looking at all LO(W0) and LO(W2) combinations, the critical condition for higher LN regeneration was primary shoots from the node cutting (primary, 63.5%), followed by normal leaves during subculture (normal, 56.5%) and liquid overlay at week 2 {LO(W2), 56.3%} > at week 0 {LO(W0), 41.5%} and hyperhydric leaves (41.3) > secondary shoots (34.2%). 

## 3. Discussion

### 3.1. In Vitro Propagation System of Endangered Wild Species

The in vitro propagation of plant material is essential, especially for endangered species with limited source materials, to prepare explants for cryopreservation experiments and restoration in situ [5]. Nodal sections are valuable for in vitro preparation since they are genetically stable and easy to manipulate. The subculturing of donor plantlets is affected by diverse factors, i.e., source material, growth medium, culture conditions, etc. We established standard subculture conditions for *S. kakudensis*. The nodal segments were inoculated onto a hormone-free MS medium gelled with Gellan gum under one fluorescent lamp (40 µE m^−2^ s^−1^), followed by a liquid medium overlay on top of the Gellan gum-gelled medium. 

As a gelling agent, Gellan gum increased plant height (1.7-fold) and dry weight (2.4-fold) compared to agar, like other wild species [14,15,16,17]. Though the mechanism is not clearly understood, it has fewer organic substance impurities and better water availability, associated with a lower concentration (3 g L^−1^) than the agar (8 g L^−1^), thus promoting the absorption of substances and facilitating the diffusion of inhibitive molecules over agar [18,19,20,21]. The concentration of gelling agents may affect water availability and tissue water potential [22]. However, the responses of gelling agents are species-specific [23,24].

Liquid overlay on top of the Gellan gum-gelled medium significantly increased plant height (2.47–2.58-fold) and dry weight (3.41–4.15-fold) compared to the conventional gelled medium. In liquid overlay application, the composition of gelled and liquid medium (nutrients and sucrose) determines the growth; adding nutrients and sucrose in the gelled medium was critical for the length and dry weight of the subcultured plantlets. Omitting the nutrients and sucrose in a liquid medium did not significantly affect plant height and dry weight. Therefore, even a liquid overlay of distilled water on top of the gelled medium {G(+N+S)+L(−N−S)} affected height and dry weight equally to the nutrient- and sucrose-supplemented liquid medium {G(+N+S)+L(+N+S)}. 

These results support the hypothesis raised by Lee et al. [16]: the liquid overlay may facilitate metabolism by modulating the osmotic potential of gelled medium and plantlets. In the previous study, liquid overlay increased the height (2.06-fold) and dry weight (3.69-fold) of *Pogostemon yatabeanus*, which limitedly occurs in wetlands [16]. The recommended optimum stage for liquid overlay for *S. kakudensis* plantlets was two weeks after inoculation (W2). The causes may include symptoms of hyperhydric leaves noticed in W0 and W1 plantlets (Table 1, Figure 2), resulting in lower dry weight during subculture and subsequent lower LN regeneration compared to the LO(W2). The preliminary screening of other species shows that the adverse effect of W0 (liquid overlay at inoculation) can be variable and seems erratic depending on the species and their vigor, explants (number/length of nodes, with or without leaves), etc. 

Hyperhydricity often happens in plant tissue culture, but no consensus exists on its causes and measures [25]. Its causes and symptoms include the waterlogging of the apoplast (intercellular space), hypoxia-associated oxidative stress, stress-induced reactive oxygen species (ROS) and ethylene accumulation, and structural abnormalities of stomata, leading to an imbalance between the absorption and loss of water in the tissue, and less lignin content (hypolignification) [25,26,27]. In the current study, the liquid overlay at the initial stage of inoculation (W0) increased the dry weight compared to no liquid overlay. Still, it was ineffective in the subsequent LN regeneration. It causes hyperhydric leaves, possibly caused by osmotic shock due to the penetration of the liquid medium into the intercellular space [25]. Therefore, the liquid overlay days after inoculation (DAI) 14 was superior to DAI 7 or no liquid overlay on somatic embryogenesis initiation in *Pinus taeda* [28]. Further investigations are needed to standardize liquid overlay application based on the underlying mechanism.

### 3.2. Droplet-Vitrification Procedure—Protocol Development

The node-section-induced shoots propagated by standard subculture conditions were subjected to DV cryopreservation. The recovery of cryoprotected control (LNC) and cryopreserved (LN) explants was affected by all of the factors investigated, i.e., the sucrose concentration in preculture, osmoprotection, cryoprotection (PVSs), cooling device (foil/vial), and regrowth medium (ammonium ion and growth regulators) [8,12]. 

A two-step preculture, 10% sucrose for 31 h, followed by 17.5% for 17 h, produced a higher LN regeneration than the one-step 10% sucrose preculture. Though one-step preculture with 10% (0.3 M) sucrose for a few days has been dominantly applied [12], two-step preculture produced 10–23% higher LN regeneration: *Dendrathema grandiflourum* (76.3% → 86.7%) [29], *Castilleja levisecta* (59.5% → 82.5%) [30], *Aster altaicus* (47% → 69.2%) [15], and *Chrysanthemum morifolium* (18.7% → 38.8%) [31]. However, 25% sucrose as a second-step preculture was detrimental. Hence, *S. kakudensis* can be grouped in a median osmotic stress.

Among the PVSs tested, A3-80% produced 18% higher LN regeneration than PVS2 in standard procedure. A3-80%-cryoprotected shoot tips may have lower water content than the PVS2-cryoprotected shoot tips since relatively higher concentrations of sucrose and glycerol in A3-80% may facilitate dehydration (the efflux of water) and cryoprotection (the influx of CPAs) [32] and thus increase the cell viscosity to the critical point at which ice crystallization is suppressed intracellularly and extracellularly [33]. A3-80%, a dilution of A3-90%, has produced 8.5~67.3% higher LN regeneration over PVS2 [12], often combined with the two-step preculture of a higher concentration of sucrose (S-17.5%).

Considering the regeneration before (LNC, Figure 6A) and after (LN, Figure 6B) and the composition of PVSs tested, causal factors for the relatively lower LN regeneration of other PVSs seem variable: PVS2—insufficient cryoprotection, A3-90%—chemical cytotoxicity, PVS3—osmotic stress, and B5-85%—osmotic stress and partly insufficient cryoprotection. Most vitrification-based studies have tested the incubation time of PVS2 [34]. However, it is worth investigating the concentration of PVS rather than the cryoprotection duration for the normal-size shoot tips since tolerance to cytotoxicity can determine the concentration of PVS, and explant size and permeability can reflect the incubation time [12]. Based on the classification, *S. kukudensis* shoot tips are sensitive to osmotic stress and median sensitive to the biochemical cytotoxicity of PVS [12].

As a cooling device, cryovial (vitrification method) produced 30% lower LN regeneration than aluminum foil strips (DV method), implying insufficient cryoprotection with A3-80%, resulting in freezing injury when cooling velocity was not sufficiently high [12]. Rapid warming is critical when the vitrified state is metastable and vulnerable to devitrification. In this regard, thawing the cryopreserved samples into a pre-heated (40 °C) unloading solution is superior to a room-temperature solution [35]. 

In this study, the most critical conditions for LN regeneration are the regrowth medium: initially omitting ammonium ions and adding growth hormones (GA_3_ + BA) for 5 days in step 1. However, including or excluding growth hormones in steps 2 and 3 was not critical. Plant growth hormones play a vital role in the regeneration of cryopreserved plant materials [36,37], and GA_3_ combined with cytokinins usually promotes direct plant formation from cryopreserved shoot tips [38]. The optimum regrowth medium and exogenous additives may vary depending on the material’s needs and injuries during the cryopreservation procedure [39]. 

During the multi-stage vitrification procedure, shoot tips are susceptible to cumulative injuries, such as osmotic stress, biochemical cytotoxicity, freezing injuries, and oxidative stress [12]. The injuries peak at the cooling, rewarming, and unloading stage, where metabolic rates are the lowest [40], while total organic acid contents jump [41]. In this case, ammonium ions (NH_4_^+^) in the regrowth medium might cause ROS-induced oxidative stress [42,43,44]. Hence, the initial ammonium-free medium increased LN regeneration over the ammonium-containing medium: *Ipomoea batatas* (by 61% [45]), *Holostemma annulare* (by 26–36% [46] and by 46% [47]), orchid protocorms (by 34% [48]), *C. morifolium* (by 36–38% [49] and by 69% [31]), *Citrus limon* (by 17% [50]), *A. altaicus* (by 33% [15]), *P. yatabeanus* (by 73% [13]), *and Penthorum chinense* (by 32% [17]). A three-step regrowth medium, i.e., initially ammonium-free and with growth hormones → ammonium-containing → hormone-free, has been recommended for the sensitive species [39]. 

### 3.3. Droplet-Vitrification Procedure—Liquid Overlay-Induced Donor Plant Vigor

Since the liquid overlay on top of the Gellan gum-gelled medium significantly increased donor plantlets’ height and dry weight, and the liquid overlay timing and composition determined the growth, we investigated the effect of the liquid overlay on LN regeneration using the five variant conditions selected (Figure 8). 

The liquid overlay of MS liquid medium at week two {LO(W2)} produced the highest LN regeneration (80.0%), followed, in order, by the liquid overlay of MS liquid medium at week 0 {LO(W0), 64.2%} > liquid overlay of distilled water {LO(W2), G(+N+S)+L(−N−S), 63.3%} > no liquid overlay {LX, 56.8%}. In LO(W0), symptoms of hyperhydricity were noticeable in the leaves, with eventually lower LN regeneration, even though only the normal plantlets were node-sectioned for cryopreservation. In this study, LN regeneration is highly correlated with shoot height and shoot dry weight of in vitro subcultured plantlets, implying that donor plant vigor is a determining factor for cryopreservation. 

In further investigation using the three conditions (WO, W2, and LX) during the subculture, we observed hyperhydricity in the leaves with WO (31.3%) and W2 (4.2%) (Table 1). These normal and hyperhydric plantlets were separately subjected to node sectioning for shoot-tip sampling for cryopreservation. Node sections usually produced two axillary shoots, and one (primary) grew faster than the other (secondary). The positive (higher dry weight) and adverse (hyperhydric leaves) effects are reflected in the regeneration of LNC and LN shoot tips. We hypothesize that the critical factors for cryopreservation are tolerance to osmotic stress and cytotoxicity before cryo-exposure. With all combinations tested, the normal plantlets during the subculture (56.5%) and primary shoots from the node section (63.5%) produced higher LN regeneration. In contrast, hyperhydricity (41.3%) and secondary shoots (34.2%) resulted in lower LN regeneration. This result suggests that donor plant vigor is determined by liquid overlay during the subculture and by selecting primary shoots from the node section during explant extraction. 

This study highlights the importance of liquid over-induced donor plant vigor and selection of dominant shoots. Like primary dominance in this result, apical dominance over axillary shoots has been broadly recognized in plant cryopreservation [51]. The preconditioning of donor plants (cold acclimation) [52] and selecting smaller explants (0.5–1.0 mm) from young shoots or plantlets [53,54] affected LN regeneration. Subculture conditions (aerating culture vessels, sparse planting density, and higher light intensity) helped produce healthy donor plants, resulting in a higher recovery of LNC and LN potato shoot tips [9]. Diverse and integrated approaches are needed to improve the donor plant vigor of plant material, especially scarce and endangered species. 

## 4. Materials and Methods

### 4.1. Plant Material, In Vitro Establishment and Preparation of Donor Plants

A few *S. kakudensis* seeds were introduced from the Gyeonggi Forest Environment Research Institute, Osan-si, Republic of Korea. In vitro plants were germinated from the seeds following a sterilization procedure using a 1.25% (*v*/*v*) NaOCl solution for 10 min, followed by thorough rinsing with sterile distilled water. Developed plants were propagated using single nodal segments (1.0~1.5 cm long) via repeated (sequential) subcultures with hormone-free MS medium (MSF, [55]) with 30 g L^−1^ sucrose and 3.0 g L^−1^ Gellan gum (MB Cell, Seoul, Republic of Korea) in 300 mL Gaooze^TM^ culture vessels (Korea Scientific Technology Industry, Suwon, Republic of Korea). The medium pH was adjusted to 5.8 before autoclaving at 121 °C for 25 min. Six single nodal segments were placed per vessel and kept at 25±1 °C under a 16/8 h light/dark photoperiod and a 40 W fluorescent lamp (40 µE m^−2^ s^−1^) for 6 weeks. 

Liquid MSF medium (15 mL per vessel) was added on top of Gellan gum-gelled medium at around day 14 of each subculture cycle, referred to as LO(W2), as the standard condition to promote the robust growth and development of the plants. After the liquid overlay, the lids of the culture vessel were sealed with cling tape to protect the liquid from evaporation. 

### 4.2. Effect of Subculture Conditions in In Vitro Propagation of Scrophularia kakudensis

Ten combinations of in vitro subculture medium and conditions were investigated to establish the in vitro propagation system. As outlined in Table 2, the combinations involved variations in MS medium strength, growth regulators, gelling agents, activated charcoal, liquid overlay medium application timing, and light intensity. The growth medium and conditions were the same as in Section 4.1. The experiment included two replicates for each treatment, and the entire set of experiments was repeated twice to ensure reliability. 

After six weeks, the dry weight (mg) and height (cm) of shoots and roots were measured separately. The dry weight was determined by subjecting the plant samples to an oven at a temperature of 75 °C for 7 h until a consistent weight was attained.

To investigate the effect of the composition of Gellan gum-gelled medium and liquid overlay medium, nodal sections were subcultured using eight culture media, each adding (+) or excluding (−)full-strength MSF nutrients and sucrose (Table 3). The other conditions were the same as in Section 4.1. 

### 4.3. Experimental Design of Treatments in the Droplet-Vitrification (DV) Procedure

#### 4.3.1. Standard Droplet-Vitrification Procedure

After repeated standard subcultures, shoot tips (1.5 mm long with 1–2 lateral leaves) were excised from 4-week-old node sections cultured on standard subculture media. A DV procedure was adopted by Lee et al. [31]. Briefly, shoot tips were precultured in 10% sucrose (S-10%) for 31 h and 17.5% sucrose (S-17.5%) for 16 h at room temperature (RT), osmoprotected with C4-35% (17.5% glycerol + 17.5% sucrose, *w*/*v*) for 30 min at RT, and then cryoprotected with PVS A3-80% (29.2% glycerol + 11.7% dimethyl sulfoxide + 11.7% ethylene glycol + 17.4% sucrose, *w*/*v*) for 60 min at 0 °C. Cryoprotected shoot tips were positioned onto aluminum foil strips (7 × 20 mm × 50 µm) containing 5 µL droplets of ice-cold A3-80% and subsequently plunged into liquid nitrogen (LN) for a minimum of 1 h. For rewarming and unloading, foil strips with shoot tips were transferred to 20 mL pre-heated (40 °C) unloading solution of 35% sucrose (S-35%) and kept for 40 min at RT, with the sucrose solution being replaced after the first 15 min. The same procedure was followed for cryoprotected control (LNC) shoot tips, excluding the cooling and rewarming in LN. 

The shoot tips retrieved from S-35% were carefully blotted to remove moisturize on sterilized filter paper and transferred to regrowth medium 1 {RM1, NH_4_NO_3_–free MS medium + 1 mg L^−1^ gibberellic acid (GA_3_) + 1 mg L^−1^ benzyl adenine (BA), 30 g L^−1^ sucrose, 3.0 g L^−1^ Gellan gum, pH 5.8} and cultured in the dark at 25 °C. After five days, explants were transferred to the RM2 medium (the same as RM1 except for NH_4_NO_3_) and cultured under 40 µE m^−2^s^−1^ for 23 days. The developed shoots were transferred to the MSF medium for 14 days for normal regeneration. 

#### 4.3.2. Sets of Experimental Conditions—Pre-LN Stages, Three-Step Regrowth, and Liquid Overlay 

Diverse variants, such as pre-LN stages, three-step regrowth, and liquid overlay conditions, were tested in the DV procedure (Table 4). During the experiment for each factor, other conditions and steps in the procedure remained constant with the standard condition (indicated as “st” in Table 4). 

We devised sets of treatments to optimize the droplet-vitrification protocol. Pre-LN stages include two-step or one-step preculture, with or without osmoprotection, cooling/warming device (aluminum foil strips vs. cryovial), and regrowth medium (ammonium-free/containing, transfer/non-transfer). When we used the cryovial (so-called vitrification method), cryoprotected shoot tips were transferred to 2 mL cryovials (Nalgene Co., Rochester, NY, USA) containing 0.5 mL of ice-cold A3-80% solution. These cryovials were immersed in LN for 1 h and rewarmed in a water bath pre-heated at 40 °C, followed by unloading and regrowth under the standard procedure. The original PVS2 and PVS3 and their variants A3-90%, A3-80%, and B5-85% were compared.

In the post-LN stage, seven combinational treatments of steps 1 (ammonium-free/containing, with/without growth regulators) and 2 and 3 (with/without growth regulators) were investigated. 

Five combination treatments were designed based on the results of Table 2 and Table 3 to elucidate the effect of liquid overlay-induced donor plant vigor. These include with/without, the timing of liquid overlay, and the composition of gelled and liquid medium (nutrients and sucrose). The more intensive investigation of three liquid overlay treatments (W0, W2, and LX) was designed for sampling in the last node section cultures. 

### 4.4. Recovery Assessment and Statistical Analysis

Survival was evaluated two weeks following cryopreservation by counting the number of shoot tips showing a regrowth of greenish tissues. Regeneration was determined after six weeks when the shoots had developed into normal plantlets (≥10 mm) with fully expanded leaves and roots, without either the lag phase or callus formation. Ten to twelve shoot tips were used per experimental condition, and the experiments were replicated no less than twice. 

Data from all experiments were analyzed by an analysis of variance (ANOVA) and least significant difference (LSD) test or Duncan’s multiple range test (DMRT, *p* < 0.05) using SAS (SAS Institute Inc., Cary, NC, USA). Results are presented as percentages with their standard deviations.

## 5. Conclusions

We developed an in vitro propagation system for *Scrophularia kakudensis* with an overlay of a liquid medium onto the Gellan gum-gelled medium. Liquid overlay at week two significantly increased the plant height (2.5-fold) and dry weight (4-fold) of the subcultured plantlets compared to the conventional no-liquid overlay. This increased the impact of donor plant vigor on the regeneration of cryopreserved (LN) shoot tips by 23.2%. 

The variation in the response of diverse plant species to cryo-exposure has been considered one of the main barriers to the cryobanking of wild flora. However, this study coincides with the previous studies on endangered wild species in the protocol developed using a systematic approach and proposed by the authors: a liquid overlay on Gellan gum-gelled medium during subculture, two-step preculture using 17.5% sucrose, cryoprotection with alternative PVS A3-80%, and three-step regrowth, initially without ammonium and with growth regulators. This optimum condition for the droplet-vitrification procedure resulted in 80% LN regeneration. This systematic approach has proven repeatable and reliable for wild species. 

We further investigated the timing and composition of the liquid overlay on the growing of the donor plants during the last subculture, as well as sprouted shoots from the nodal sections during the sampling stage. The main findings imply that the donor plant vigor is the most critical factor for higher LN regeneration; overlay the liquid medium on top of the gelled medium at week two to prevent the hyperhydricity of leaves during the subculture and select primary shoots from the node section for explant extraction. This study highlights the usefulness of liquid overlay onto the Gellan gum-gelled medium in the vigorous growing of in vitro plantlets during the subculture and subsequent high regeneration of cryopreserved shoot tips. 

## Figures and Tables

**Figure 1 plants-13-02408-f001:**
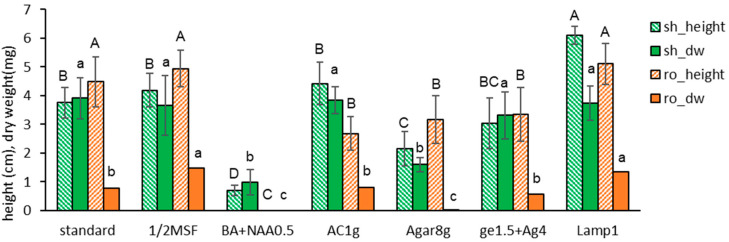
Plant height (cm) and dry weight (mg) of in vitro grown *Scrophularia kakudensis* plantlets with seven treatments of subculture medium and conditions. standard—growth hormone-free MSF medium with 30 g L^−1^ sucrose and 3.0 g L^−1^ Gellan gum, two lamps (60 µE m^−2^ s^−1^); 1/2MSF—same medium with half-strength macro mineral salts; BA + NAA0.5—MS medium + 0.5 mg L^−1^ benzyl adenine (BA) and 0.5 mg L^−1^ α-naphthaleneacetic acid (NAA); AC1g—activated charcoal 1.0 g L^−1^; Agar8g—agar 8 g L^−1^; and Lamp1—illumination of 40 µE m^−2^ s^−1^ provided by one fluorescent lamp. Means with the same letters (A–D and a–c) in each column are not significantly different by least significant difference test (LSDT, *p* < 0.05).

**Figure 2 plants-13-02408-f002:**
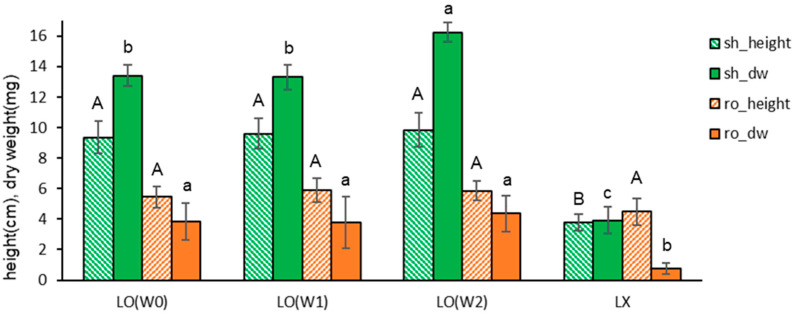
Plant height (cm) and dry weight (mg) of in vitro grown *Scrophularia kakudensis* plantlets with the timing of liquid overlay. LO— an overlay of the liquid medium on top of the gelled solid medium at the time of inoculation (W0), one week (W1) after, or two weeks (W2) after. Liq-X—no liquid overlay. Means with the same letters (A,B and a–c) in each graph are not significantly different by least significant difference test (*p* < 0.05).

**Figure 4 plants-13-02408-f004:**
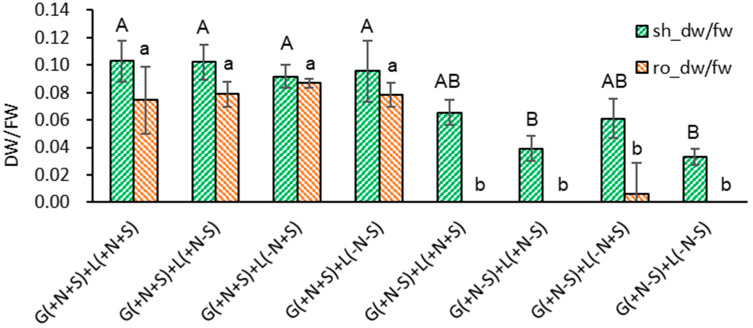
Effect of gelled medium and liquid overlay components on dry weight to fresh weight ratio (DW/FW) of subcultured *Scrophularia kakudensis* plantlets. G, Gellan gum gelled; L, liquid overlay; and inclusion (+) or exclusion (−) of nutrients (N) and sucrose (S). Means with the same letters (A,B and a,b) in each column are not significantly different by least significant difference test (LSDT, *p* < 0.05).

**Figure 5 plants-13-02408-f005:**
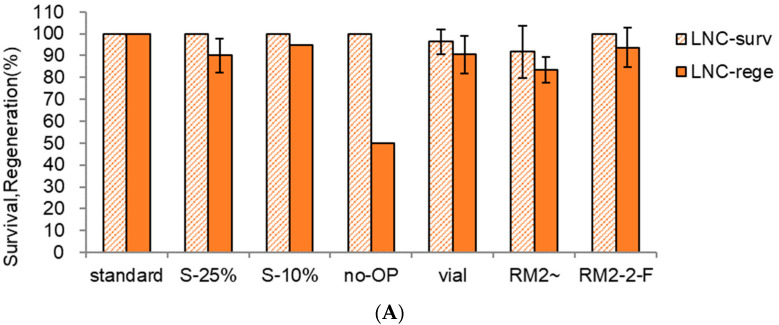
(**A**) Effect of preculture, osmoprotection, cooling and rewarming device, and regrowth medium on LNC-survival (surv) and regeneration (rege) of *Scrophularia kukudensis* shoot tips in regrowth steps of RM1 (ammonium-free)-RM2-MSF. standard—S-10% 31 h → S-17.5% 17 h, C4-35% 40 min, A3-80% ice 60 min, cooling and rewarming using aluminum foil strips, three-step regrowth RM1-2-F {RM1 (MS + NH_4_NO_3_-free + GA1 + BA0.5), 5d, dark → RM2 (MS + NH_4_NO_3_-containing + GA1 + BA0.5), 3w2d, one lamp → MSF (MS + growth hormones-free)}; S-25%, S-10% → S-25% 17 h; no-OP, no osmoprotection; vial, 2 mL cryovial with 0.5 mL A3-80%; RM2~, RM2 (MS + NH_4_NO_3_-containing + GA1 + BA0.5) without transfer to a new medium for six weeks.; RM2-2-F, RM2-RM2-MSF; GA1 + BA0.5, 1 mg L^−1^ gibberellic acid (GA_3_) + 1 mg L^−1^ benzyl adenine (BA). (**B**) Effect of preculture, osmoprotection, cooling and rewarming device, and regrowth medium on LN-survival (surv) and regeneration (rege) of *Scrophularia kukudensis* shoot tips in regrowth steps of RM1 (ammonium-free)-RM2-MSF. Means with the same letters (A,B and a–c) in each graph are not significantly different by least significant difference test (*p* < 0.05).

**Figure 6 plants-13-02408-f006:**
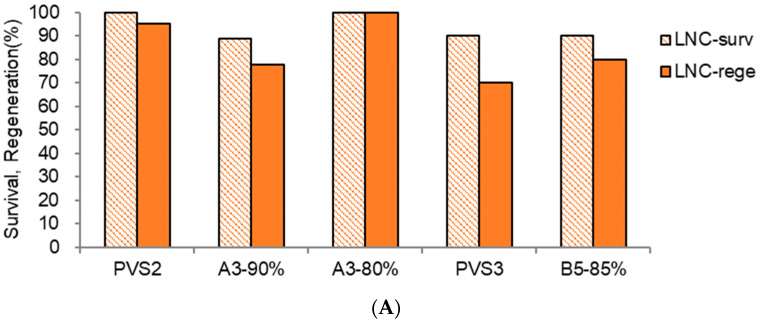
(**A**). Effect of plant vitrification solutions on LNC survival (surv) and regeneration (rege) of *Scrophularia kukudensis* shoot tips in regrowth steps of RM1 (ammonium-free)-RM2-MSF. Cryoprotection: four-component PVS PVS2, A3-90%, and A3-80% for 60 min at 0 °C and two-component PVS PVS3 and B5-85% for 60 min at 25 °C. (**B**). Effect of plant vitrification solutions on LN-survival (surv) and regeneration (rege) of *Scrophularia kukudensis* shoot tips in regrowth steps of RM1 (ammonium-free)-RM2-MSF. Means with the same letters (A and a,b) in each graph are not significantly different by least significant difference test (*p* < 0.01).

**Figure 7 plants-13-02408-f007:**
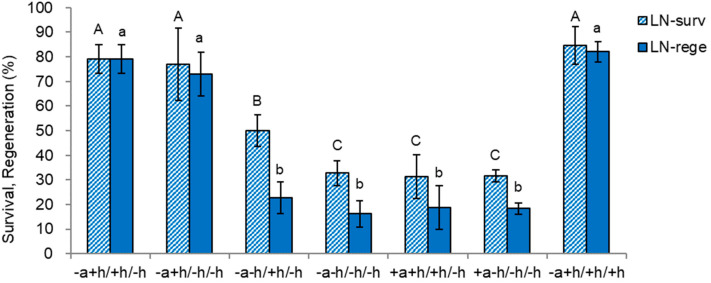
Effect with or without ammonium nitrate and growth regulators at three regrowth steps on survival and regeneration of cryopreserved (LN) *Scrophularia kukudensis* shoot tips. step 1/step 2/step 3. Step 1 was performed on MS medium with (+) or without (−) ammonium nitrate and growth regulators (1 mg L^−1^ GA_3_ + 0.5 mg L^−1^ BA) in the dark for five days. Steps 2 and 3 were performed on MS medium containing ammonium nitrate with (+) or without (−) 1 mg L^−1^ GA_3_ + 0.5 mg L^−1^ BA under light, and 40 µE m^−2^ s^−1^ for 23 and 14 days, respectively. Means with the same letters (A–C and a,b) in each graph are not significantly different by least significant difference test (*p* < 0.05).

**Figure 8 plants-13-02408-f008:**
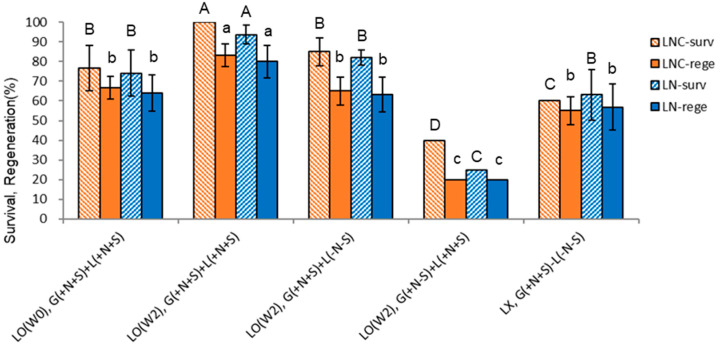
Effect of liquid overlay and timing {LO(W0), LO(W2), LX} and composition of Gellan gum-gelled medium and liquid medium on survival (surv) and regeneration (rege) of cryoprotected (LNC) and cryopreserved (LN) *Scrophularia kukudensis* shoot tips. Liquid overlay (LO) at the time of inoculation (W0), two weeks (W2) after inoculation, or no liquid overlay (LX). The composition of gelled and liquid medium includes (+) or excludes (−) nutrients (N) and sucrose (S). Means with the same letters (A–D and a–c) in each graph are not significantly different by least significant difference test (*p* < 0.05).

**Figure 9 plants-13-02408-f009:**
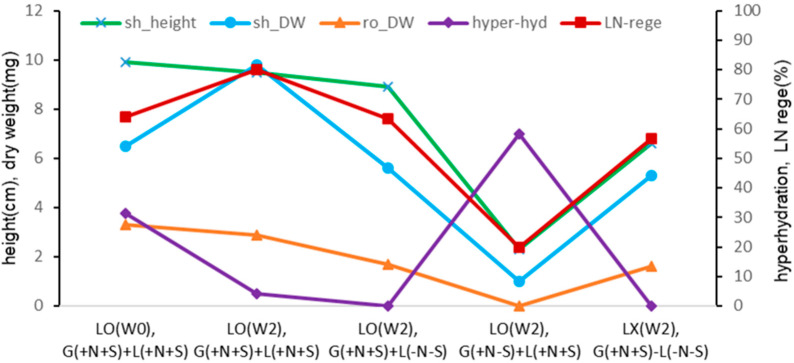
Effect of liquid overlay and timing and composition of Gellan gum-gelled medium and liquid medium on plant height, dry weight, hyperhydricity, and cryopreserved (LN) regeneration in *Scrophularia kukudensis* shoot tips.

**Figure 10 plants-13-02408-f010:**
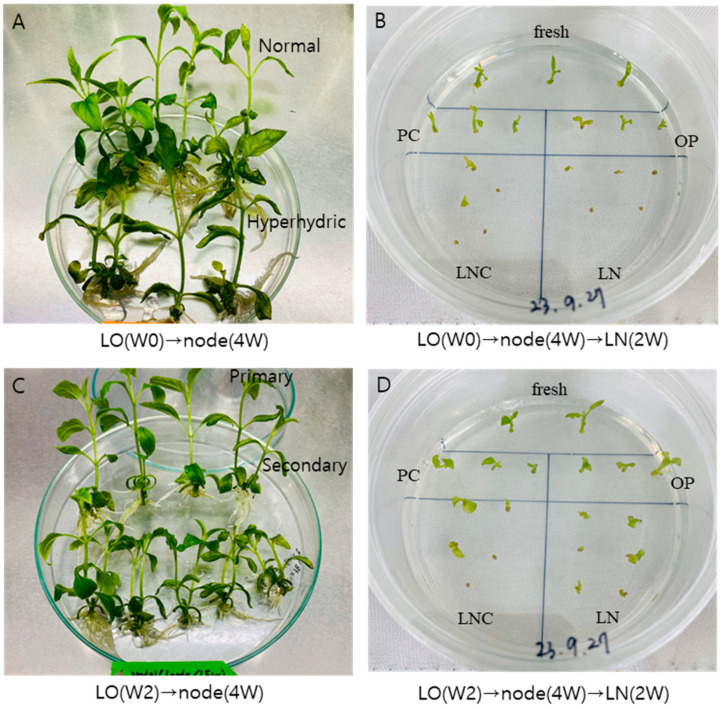
Preparation of node section-induced primary and secondary shoots from liquid overlay at the time of node inoculation {LO(W0), (**A**,**B**)} and week 2 {LO(W2), (**C**,**D**)}. Liquid overlay subcultured plantlets were node section cultures for 4 weeks (**A**,**C**) followed by droplet-vitrification cryopreservation using LO(W0) → normal-primary shoots (**B**) and LO(W2) → normal-primary shoots (**D**) and regrowth for 2 weeks using ammonium-free regrowth medium 1 (RM1)-RM2 in *Scrophularia kakudensis*. (**A**) Hyperhydric shoots are watery and look darker than normal shoots. (**B**,**D**) Overall, the shoot tips for all stages of LO(W2) (**D**) grow faster than LO(W0) (**B**); in particular, the LN shoot tips for LO(W2) noticeably grow faster than LO(W0). Fresh, fresh control; PC, precultured; OP, osmoprotected; LNC, cryoprotected but not cryopreserved; LN, cryopreserved.

**Table 1 plants-13-02408-t001:** Effect of liquid overlay (timing and composition) in the last subculture cycle on the hyperhydricity of leaves, sprouting of node section-induced shoots, and subsequent LNC and LN regrowth in *Scrophularia kukudensis*.

No.	Normal/Hyperhydricity in Subcultured Plantlets (%)	Nodal Segment-Induced Shoots		LNC (%)		LN (%)	
		Primary/Secondary (cm) **	Hyperhydricity (%) ***	Surv	Rege	Surv	Rege
1. LO(W0) *	Normal (68.7%)	Primary (4.5)	41.7	76.7	66.7	74.2	64.2
		Secondary (2.0)	91.7	60.0	50.0	38.3	31.7
	Hyperhydric (31.3%)	Primary (4.5)	100	80.0	50.0	60.0	50.0
		Secondary (1.5)	100	60.0	30.0	30.0	20.0
4. LO(W2)	Normal (95.8%)	Primary (6.5)	0	100.0	83.3	93.8	80.0
		Secondary (2.5)	25	70.0	60.0	55.0	50.0
	Hyperhydric (4.2%)	Primary (6.5)	100	90.0	60.0	75.0	60.0
		Secondary (1.5)	100	70.0	40.0	45.0	35.0
7. LX	Normal (100%)	Primary (4.0)	0	60.0	55.0	62.1	56.8
		Secondary (2.0)	0	50.0	30.0	40.0	25.0
	Hyperhydric (0%)	-	-	-	-	-	-

* Liquid overlay (LO) at the time of inoculation (W0), two weeks after inoculation (W2), or no liquid overlay (LX). After 6 weeks of subculture, the plantlets were separated into normal and hyperhydricity and subjected to final node sectioning for a sampling of cryopreservation. ** The length (cm) of primary and secondary node-induced shoots at week 4 of node-segment culture. *** Percentage of hyperhydricity symptoms in node segment-induced shoots after 4 weeks.

**Table 2 plants-13-02408-t002:** Set of treatments to test the impact of subculture medium and conditions for 6 weeks on the growth of donor plantlets and regrowth of cryopreserved *Scrophularia kakudensis* shoot tips.

Set	No.	Medium Strength	Growth Regulators (mg L^−1^)	Gelling Agents (g L^−1^) **	AC (g L^−1^) ***	Liquid Overlay ****	Light (Lamp) *****	Code
Subculture medium and conditions	1	MS	−	Gellan gum 3	−	LX	2	MSF *, standard
	2	1/2MS	−	Gellan gum 3	−	LX	2	1/2MSF
	3	MS	BA0.5 + NAA0.5	Gellan gum 3	−	LX	2	B0.5N0.5
	4	MS	−	Gellan gum 3	1	LX	2	AC1
	5	MS	−	Agar 8	−	LX	2	Agar8
	6	MS	−	Gell 1.5 + Agar 4	−	LX	2	Gel1.5 + Agar4
	7	MS	−	Gellan gum 3	−	LX	1	L1
Timing of liquid overlay	1	MS	−	Gellan gum 3	−	LX	2	LX
	2	MS	−	Gellan gum 3	−	LO(W0)	2	LO(W0)
	3	MS	−	Gellan gum 3	−	LO(W1)	2	LO(W1)
	4	MS	−	Gellan gum 3	−	LO(W2)	2	LO(W2)

* MSF, standard hormone-free MS (MSF) medium with 30 g L^−1^ sucrose and 3.0 g L^−1^ Gellan gum without liquid overlay; 1/2MS, half-strength MSF medium; −, omission; growth regulators—B0.5 + N0.5, {0.5 mg L^−1^ benzyl adenine (BA) + 0.5 mg L^−1^ α-naphthaleneacetic acid (NAA)}; ** gelling agent—{3 g L^−1^ Gellan gum (standard), 8 g L^−1^ agar (Agar8), 1.5 g L^−1^ Gellan gum and 4 g L^−1^ agar (Gel1.5+Agar4)}; *** AC1—activated charcoal 1.0 g L^−1^; **** liquid overlay—overlay (LO) at week 0 (W0), week 1 (W1), week 2 (W2), no overlay (LX); ***** light—one or two lamps providing 40 and 60 µE m^−2^ s^−1^, respectively).

**Table 3 plants-13-02408-t003:** Set of treatments to test the impact of subculture medium nutrients (N) and sucrose (S) on the growth of donor plantlets and regrowth of subsequently cryopreserved *Scrophularia kakudensis* shoot tips.

No.	Gellan Gum-Gelled Medium *	Liquid-Overlay Medium **	Code
A1 ***	G(+N, +S)	+L(+N, +S)	G(+N+S)+L(+N+S)
A2	G(+N, +S)	+L(+N, −S)	G(+N+S)+L(+N−S)
A3	G(+N, +S)	+L(−N, +S)	G(+N+S)+L(−N+S)
A4	G(+N, +S)	+L(−N, −S), ddw	G(+N+S)+L(−N−S)
B1	G(+N, −S)	+L(+N, +S)	G(+N−S)+L(+N+S)
B2	G(+N, −S)	+L(+N, −S)	G(+N−S)+L(+N−S)
B3	G(+N, −S)	+L(−N, +S)	G(+N−S)+L(−N+S)
B4	G(+N, −S)	+L(−N, −S)	G(+N−S)+L(−N−S)

* G, Gellan gum gelled; ** L, liquid overlay; adding (+) or excluding (−)full-strength MS nutrients (macro, micro, amino acids, vitamins, and N) and sucrose (30 g L^−1^, S). *** A1, as the standard condition, corresponds to ‘4. LO(W2)’ in Table 1.

**Table 4 plants-13-02408-t004:** Set of treatments to test the impact of pre-LN, LN, and post-LN procedures on the growth of cryopreserved *Scrophularia kukudensis* shoot tips.

Procedure		Treatment Conditions			Code
Protocol	Preculture	10% sucrose, 31 h → 17.5% sucrose, 16 h			S-17.5%, st ***
		10% sucrose, 31 h → 25% sucrose, 16 h			S-25%
		10% sucrose, 48 h			S-10%
	Osmoprotection	C4-35%, 30 min			OP, st
		No osmoprotectant			no-OP
	Cooling/warming device	Aluminum foil-strips			Foil, st
		Cryovial (2 mL)			Vial
	Regrowth medium *	RM1-RM2-MSF			RM1-2-F, st
		RM2~			RM2~
		RM2-RM2-MSF			RM2-2-F
	Cryoprotection **	A1-73.7% (PVS2) ice, 60 min			PVS2
		A3-90% ice, 60 min			A3-90%
		A3-80% ice, 60 min			A3-80%, st
		B1-100 (PVS3) 25 °C, 60 min			PVS3
		B5-85% 25 °C, 60 min			B5-85%
Post-LN	Ammonium ion and growth hormones in regrowth medium	Regrowth Step 1	Step 2	Step 3	
		1. NH_4_NO_3_-free + GA1 + BA1	→GA1 + BA1	→MSF **	−a+h/+h/−h, st
		2. NH_4_NO_3_-free + GA1 + BA1	→MSF	→MSF	−a+h/−h/−h
		3. NH_4_NO_3_-free + MSF	→GA1 + BA1	→MSF	−a−h/+h/−h
		4. NH_4_NO_3_-free + MSF	→MSF	→MSF	−a−h/−h/−h
		5. NH_4_NO_3_-containing + GA1 + BA1	→GA1 + BA1	→MSF	+a+h/+h/−h
		6. NH_4_NO_3_-containing + MSF	→MSF	→MSF	+a-h/−h/−h
		7. H_4_NO_3_-free + GA1 + BA1	→GA1 + BA1	→GA1 + BA1	−a+h/+h/+h
Donor plant vigor	Liquid overlay	1. Liquid overlay at week 0, G(+N+S)+L(+N+S)			LO(W0), G(+N+S)+L(+N+S)
		2. Liquid overlay at week 2, G(+N+S)+L(+N+S)			LO(W2), G(+N+S)+L(+N+S), st
		3. Liquid overlay at week 2, G(+N+S)+L(−N−S) (ddw)			LO(W2), G(+N+S)+L(−N−S)
		4. Liquid overlay at week 2, G(+N−S)+L(+N+S)			LO(W2), G(+N−S)+L(+N+S)
		5. no Liquid overlay, G(+N+S)−L(−N−S)			LX, G(+N+S)−L(−N−S)

* RM1 (MS + NH_4_NO_3_-free + GA1+BA0.5), 5d, dark → RM2 (MS + NH_4_NO_3_-containing + GA1+BA0.5), 3w2d, 1 L → MSF (MS + growth hormones-free), 2w, 2 L; GA1+BA0.5, 1 mg L^−1^ gibberellic acid (GA_3_) + 0.5 mg L^−1^ benzyl adenine (BA); 1 L and 2 L, light provided by one and two fluorescent lamps (40 and 60 µE m^−2^ s^−1^, respectively). ** A1-73.7% (PVS2), 30% glycerol + 15% DMSO + 15% EG + 13.7% sucrose, *w*/*v*; A3-90%, 37.5% glycerol + 15% DMSO + 15% EG + 22.5% sucrose, *w*/*v*; A3-80%, 33.3% glycerol + 13.3% DMSO + 13.3% EG + 20.1% sucrose, *w*/*v*; B1-100% (PVS3), 50% glycerol + 50% sucrose, *w*/*v*; B5-85%, 42.5% glycerol + 42.5% sucrose, *w*/*v*; C4-35%, 17.5% glycerol + 17.5% sucrose, *w*/*v*; DMSO, dimethyl sulfoxide; EG, ethylene glycol. *** st, standard indicates treatments composing the standard procedure where other stages are the same as in the standard protocol.

## Data Availability

The original contributions presented in the study are included in the article, further inquiries can be directed to the corresponding authors.

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
