# Peer review of "Liquid Overlay-Induced Donor Plant Vigor and Initial Ammonium-Free Regrowth Medium Are Critical to the Cryopreservation of Scrophularia kakudensis"

_plants, 2024, doi:10.3390/plants13172408_

Round 1

Reviewer 1 Report

Comments and Suggestions for Authors

In this study, an attempt was made to develop a method for the in vitro culture and cryopreservation of shoot tips of Scrophularia kakudensis, a valuable medicinal plant. At the same time, the crucial role of the usefulness of a liquid layer on the gellant gum medium for the vigorous growth of in vitro plantlets during subculture and the subsequent high regeneration of the cryopreserved shoot tips was investigated. Creating the basis for in vitro propagation and low temperature preservation of endangered medicinal plants.

The manuscript follows the structure recommended by Plants. The research design and methods used are appropriate to achieve the purpose of the study. The results are well presented and supported with high quality tables and figures.

The topic covered is interesting, but there are some ambiguities in the manuscript.

1.       Results

2.2.1 Pre-LN stages in the droplet vitrification process

The comments on Figures 6 and 8 lack a description of the significance of the difference.

2.        

Discussion

3.2 Droplet vitrification process – development of the protocol

Is there a literature reference to the influencing factors in the first paragraph? If so, it is suggested that the literature be supplemented.

3.        

Materials and Methods

4.1 Plant material, in vitro establishment and preparation of donor plants

The suggestions for the seed sources in the plant material is clearly formulated.

4.2 Effect of subculture conditions in the in vitro propagation of Scrophularia kakudensis

What does 'A1 corresponds to LO(W2) in Table 1' mean in the comments in Table 3? Does A2~B4 correspond?

4.3.2 Sets of experimental conditions - pre-LN stages, three-stage regrowth and liquid overlay

The comment in Table 4 suggests that solution components such as A1-73.7% (PVS2) and A3-90% match the RM1 solution components.       

4       References

Please reconfirm the formatting of the reference section. There are few sources in the references in recent years.

Author Response

Thank you very much for taking the time to review this manuscript. Please find the re-submitted file to see all the changes corresponding to your comments. 

Reviewer 2 Report

Comments and Suggestions for Authors

Although the experimental wrl is very fine but the authors have failed to mention the aim of the study. It should be given as last paragraph of the Introduction.

Author Response

Thank you very much for taking the time to review this manuscript.

The changes are in the revised file. We changed some words in the last sentence of the Introduction.

Reviewer 3 Report

Comments and Suggestions for Authors

I have read with great interest the manuscript "Liquid overlay-induced donor plant vigor and initial ammonium-free regrowth medium are critical to the cryopreservation of Scrophularia kakudensis ". The objective of this paper was to establish an in vitro propagation system for S. kakudensis and develop a cryopreservation protocol using the droplet-vitrification technique. It is an interesting contribution for this area of research. My main concerns are related to the figures and tables, which should be carefully revised. I can nevertheless see and value the enormous amount of work invested in this paper, and I am therefore suggesting the authors to critical revise it. Furthermore, some queries are also needed to be clarified, as indicated in the following detailed comments. 

Detailed comments:

1. General considerations about the figures: all figures should be revised. Almost all of them lack some statistical data represented by the lowercase or upper-case letters. For example: in figures 1 and 2, there are no statistical analysis regarding root height (length?) and dry matter.

2. Figures 5 and 6 should be grouped together as one figure with two line graphics, which should be cited as Figure 5-A and Figure 5-B. It would facilitate the comparison between cryopreserved and non-cryopreserved materials submitted to the same conditions. The same structure should be adopted for figures 7 and 8.

3. In the Results section, the authors related the low regeneration rates to “stressful conditions” or “ammonium-induced ROS-mediated oxidative stress during regrowth stages”. Although oxidative stress has been considered an important mechanism underlying cryoinjuries and, thus, low post-freezing recovery, the authors did not evaluate, directly or indirectly, the oxidative stress by ROS dosage, quantification of the activity of antioxidant enzymes and/or the by-products of their interaction with cellular components, such as MDA. Therefore, the affirmations described above should be carefully revised.

4. There is a Figure 11 (line graphic) in page 9 that was not cited in the text. However, there is another Figure 11 (pictures) in the next page, which is properly cited.

5. The pictures in Figure 11 (page 10) should be reconsidered. It is not possible to notice the morphological differences between normal and hyperhydric plants. The picture is too small, and the plants are covering each other. Additionally, authors should avoid the written details in the pictures B and D.

6. One of the parameters evaluated for the establishment of the cryopreservation protocol was the “cooling container”. However, the authors did not test different types of cryovials. They actually evaluated two cryopreservation techniques: vitrification and droplet-vitrification, which use distinct ways of immersion the plant materials into LN. It must be corrected in the text, since the observed effects were not related to the container, but to the several specific conditions associated with the droplet-vitrification technique, including amount of cryoprotectant solution, and cooling and rewarming rates due to the aluminum conductivity.      

7. The Discussion section must be critically revised. There are several ideas repeated from the Results, including citations of figures and tables, which should be avoided.

8. There are some minor mistakes in English expressions, especially those regarding plant tissue culture. Therefore, written English should be carefully polished.

Comments on the Quality of English Language

There are some minor mistakes in English expressions, especially those regarding plant tissue culture. Therefore, written English should be carefully polished.
